# Role of Immune Cells in Patients with Hepatitis B Virus-Related Hepatocellular Carcinoma

**DOI:** 10.3390/ijms22158011

**Published:** 2021-07-27

**Authors:** Hyo-Jung Cho, Jae-Youn Cheong

**Affiliations:** Department of Gastroenterology, Ajou University School of Medicine, Suwon 16499, Korea; pilgrim8107@hanmail.net

**Keywords:** immune cells, hepatitis B virus, hepatocellular carcinoma, innate immunity, adaptive immunity, immune balance

## Abstract

Hepatocellular carcinoma (HCC) develops almost entirely in the presence of chronic inflammation. Chronic hepatitis B virus (HBV) infection with recurrent immune-mediated liver damage ultimately leads to cirrhosis and HCC. It is widely accepted that HBV infection induces the dysfunction of the innate and adaptive immune responses that engage various immune cells. Natural killer (NK) cells are associated with early antiviral and antitumor properties. On the other hand, inflammatory cells release various cytokines and chemokines that may promote HCC tumorigenesis. Moreover, immunosuppressive cells such as regulatory T cells (Treg) and myeloid-derived suppressive cells play a critical role in hepatocarcinogenesis. HBV-specific CD8+ T cells have been identified as pivotal players in antiviral responses, whilst extremely activated CD8+ T cells induce enormous inflammatory responses, and chronic inflammation can facilitate hepatocarcinogenesis. Controlling and maintaining the balance in the immune system is an important aspect in the management of HBV-related HCC. We conducted a review of the current knowledge on the immunopathogenesis of HBV-induced inflammation and the role of such immune activation in the tumorigenesis of HCC based on the recent studies on innate and adaptive immune cell dysfunction in HBV-related HCC.

## 1. Introduction

Hepatocellular carcinoma (HCC) is one of the most aggressive cancers and is the third leading etiology of cancer-related deaths worldwide. The hepatitis B virus (HBV) is a major risk factor for the development of HCC; chronic HBV infection accounts for 70–80% of HCC cases in HBV endemic regions [1,2,3]. Persistent HBV infections generate inflammation and continuous necrosis through the immune response, which contribute to hepatocarcinogenesis.

The liver is a central immune modulator that guarantees organic and systemic preservation, while maintaining immune tolerance. It contains numerous innate and adaptive immune cells that are critical to the pathogenesis of HCC [4]. Dysregulation of the immunological network is a typical characteristic of HCC originating from chronic liver disease. A recent study showed the difference in the microenvironment between HBV-related HCC and nonviral HCC, demonstrating more immunosuppressive tumor microenvironment in HBV-related HCC [5].

Although the understanding of HBV-related HCC immuno-pathogenesis has increased, the exact roles of immune cells underlying HCC development are not yet understood. Herein, we summarized the dual roles of the innate and adaptive immune cells in patients with HBV-related HCC.

## 2. Cells of the Innate Immune Response in HBV-Related HCC

### 2.1. Natural Killer and Natural Killer T Cells

Natural killer (NK) cells are a major component of the innate immune cells in the liver, the site of HBV replication. NK cells constitute 25–50% of the total number of liver lymphocytes, which means that NK cells play a central role in innate immunity [6,7]. NK cells are associated with early antiviral and antitumor properties [8]. Inflammatory cells produce miscellaneous cytokines and chemokines that stimulate HBV-associated HCC tumorigenesis [9]. Specific blockade of interleukin (IL)-10 and transforming growth factor (TGF)-β restored the immunosuppressive milieu in patients with chronic HBV infection, via the reduction in the antiviral potency of the NK cells [10].

HBV transmission in NK cells is mediated by HBV-positive exosomes, which induce the impairment of NK cell functions, including interferon (IFN)-γ production, cytolytic activity, NK cell proliferation, and survival [11]. Moreover, the expression of microRNA-146a was increased in NK cells obtained from patients with HBV-related HCC compared to healthy individuals. NK cell cytotoxicity, and IFN-γ and TNF-α production were reduced in patients with HBV-related HCC [12].

NK cells function as a double-edged sword in chronic HBV infection [13]. NK cells have antifibrotic and/or hepatoprotective functions [14], whilst NK cell activation mediates HBV-associated hepatocyte damage [7,13]. A recent study showed that IFN-γ derived from NK cells induced HCC via the epithelial cell adhesion molecule–epithelial-to-mesenchymal transition axis in HBV transgenic mice, and hepatitis B surface antigen (HBsAg)-positive hepatocyte damage was mediated by activated NK cells, which promoted the development of HCC [15].

Natural killer T (NKT) cells derived from HBV-transgenic mice released the inflammatory cytokines IL-4 and IL-13, which promote hepatic stellate cell (HSC) activation in hepatic fibrogenesis [16]. Recently, it was shown that HSCs increased levels of T helper (Th)17 cells and upregulated regulatory T cells (Tregs), contributing to the occurrence of HCC [17].

### 2.2. Kupffer Cells and Monocytes

Kupffer cells reside in the liver sinusoids and are the largest population of immune liver cells [18]. In vivo studies have indicated that the prolonged activation of Kupffer cells and inflammatory monocytes can result in chronic liver inflammation and liver regeneration. HBs-transgenic mice with CD205-expressing Kupffer cells exhibited increased liver damage as a result of NKT cell activation via the Fas signaling pathway [19]. A key mechanism by which CD8+ T cells produce IFN-γ is that they direct macrophages to produce fibrosis-promoting cytokines and chemokines, such as TNF-α, IL-6, and MCP-1, which facilitate the progression of chronic hepatitis to liver cancer [20]. Circulating CD14+ monocytes in patients with chronic HBV infection may activate CD8+ T cells through CD137 ligand upregulation [20].

Kupffer cells also play opposing roles in HCC development during chronic HBV infection. In HBV-persistent mice, IL-10 released from Kupffer cells was essential for supporting humoral immunity [21]. Moreover, Kupffer cells supported the exhaustion of CD8+ T cells following HBV infection in mice via interactions with the hepatitis B core antigen-TLR2 [22].

### 2.3. Myeloid-Derived Suppressor Cell

The myeloid-derived suppressor cell (MDSC) is a regulatory immune cell population, which is important since it is a collection of innate immune cells residing in the liver. A previous study has demonstrated that MDSCs derived from the liver tissue of HBV transgenic mice inhibited the proliferation of allogenic T cells and lymphocytes specific to HBsAg [23]. MDSCs were also shown to play a role in both the induction and function of Tregs in patients with HCC [24]. γδT cells were found to be involved in regulating HBV-induced liver tolerance by causing MDSC infiltration in the liver, resulting in the exhaustion of MDSC-mediated CD8+ T cells [25]. Recent studies reported that Vδ2+ γδT cells from patients with chronic HBV-infection with hepatitis flares responded less to IFNγ/TNF than those without hepatitis flares [26].

## 3. Cells in the Adaptive Immune Response in HBV-Related HCC

### 3.1. CD8+ T Cells

#### 3.1.1. Effector HBV-Specific CD8+ T Cells

During acute HBV infection, effector HBV-specific CD8+ T cells produce pro-inflammatory cytokines, such as IFN-γ, IL-2, and TNF-α, and cytotoxic molecules including granzyme and perforin, to control HBV infection [27]. The activated effector T cells gradually decrease after the acute phase of HBV infection, with the resolution of infection [28]. However, various circulating HBV-specific CD8+ T cells, including HBsAg-specific, HBcAg-specific, and HBV polymerase-specific CD8+ T cells, are persistently detected in the chronic phase of HBV infection [29]. Several previous studies demonstrated that HBV-specific CD8+ T cells are critical for persistent hepatic inflammation, which ultimately leads to HCC development [30,31,32].

Persistent hepatic inflammation in chronic hepatitis B (CHB) promotes accelerated hepatocyte turnover, leads to hepatic fibrosis and cirrhosis, accumulates genetic mutations, and eventually HCC tumorigenesis [32,33,34]. The HBsAg-specific CD8+ T cell plays a crucial role in the HBV-induced hepatocarcinogenesis in various murine HBV models. Nakamoto et al. were the first to report the role of HBV-specific CD8+ T cells in HBV-induced hepatocarcinogenesis [30]. They demonstrated that the injection of splenic HBsAg-specific CD8+ T cells from HBsAg-immunized mice contributed to chronic hepatic inflammation and subsequent HCC development. They also showed that the anti-Fas ligand-neutralizing antibody, which inhibits effector T cell function, could attenuate hepatotoxicity caused by HBsAg-specific T cells, and ultimately prevent HCC development [31]. Zong et al. reported increased T cell immunoglobulin and ITIM domain (TIGIT) expression in CD8+ T cells in the immune-tolerant phase of CHB [35]. Blocking or depleting TIGIT in HBsAg-transgenic mice with an adaptive immune system that is tolerant of HBsAg led to the breakdown of adaptive immune tolerance, followed by persistent hepatic inflammation, and eventually HCC development that was dependent on CD8+ T cells [35]. A more recent study found that the depletion of CD8+ T cells completely prevented HCC development in another murine HBsAg-transgenic model. Thus, these prior studies demonstrated the critical role of CD8+ T cells in HBsAg-driven inflammation and HCC development [32].

#### 3.1.2. Exhausted HBV-Specific CD8+ T Cells

The effector HBV-specific CD8+ T cells are progressively exhausted in chronic HBV infection. The mechanisms contributing to the exhaustion of HBV-specific CD8+ T cells entail a sustained high viral load, and increased Treg cells and immune-suppressive cytokines, such as IL-10 and TGF-β [27,36,37,38]. Exhausted T cells are prone to apoptosis by the upregulation of the death receptor TRAIL-2 and the pro-apoptotic mediator BIM; they exhibit decreased proliferation and impaired effector functions (pro-inflammatory cytokine production such as IFN-γ and hepatocyte killing by perforin or granzyme) [39,40]. During the process of tumorigenesis, the exhausted CD8+ effector T cells eventually weaken the tumor surveillance of the adaptive immune system and lead to immune evasion by tumor cells. Moreover, CD8+ T cells are exhausted and an immunosuppressive environment is created in chronic HBV infection, resulting in a favorable immunological environment for tumor formation. There exists a high degree of similarity between the exhausted CD8+T cells from patients with CHB and those from patients with HBV-related HCC, with respect to the gene expression and enriched signaling pathways.

Exhausted HBV-specific CD8+ T cells characteristically express multiple co-inhibitory receptors, including the programmed cell death protein-1 (PD-1), cytotoxic T-lymphocyte antigen 4 (CTLA-4), T cell immunoglobulin and mucin domain 3 (TIM-3), and TIGIT [41,42,43,44]. The expression of these co-inhibitory receptors in HBV-specific CD8+ T cells is associated with a high viral load and is correlated with the phenotypic and functional characteristics of T cell exhaustion [44]. In HBV-related HCCs, the frequency of occurrence of CD8+ T cells with inhibitory molecules was associated with tumor aggressiveness and poor clinical outcomes. The expression of PD-1 in CD8+ T cells from HBV-related HCC was considerably higher and these cells are more functionally exhausted than CD8+ T cells from nonviral HCC [5]. HCCs with a discrete population of PD-1-high cells demonstrated a more aggressive clinical course than HCCs without a discrete population of PD-1-high cells [45]. The increased frequencies of PD-1+ TIGIT+ CD8+ T cell populations were associated with disease progression and poor prognosis in HBV-related HCC [46]. These co-inhibitory receptors, also known as immune check points, have been extensively studied in the field of cancer immunology, as well as in the development of therapeutic strategies for CHB.

In addition to a weakened immune surveillance, the exhaustion of CD8+ T cells also contributes to chronic hepatic inflammation. In the past, patients in the immune-tolerant phase were considered to be immunologically tolerant to HBV. This meant that the virus was not thought to evoke an immune response, which causes hepatic inflammation, in such patients. However, it was found that effector/inflammatory cytokines such as IFN-ɑ TNF-ɑ, IL-17A, and IL-22, which were produced by effector CD8+ T cells, were significantly higher in patients in the immune-tolerant phase than in healthy individuals [47]. It is assumed that exhausted HBV-specific CD8+ T cells are not terminally exhausted but induce chronic and significant inflammation that evokes hepatocarcinogenesis by maintaining a partially activated status with a continuous HBV antigen load.

Therefore, the exhausted CD8+ T cell maintains a partially activated state against the persistent viral antigen load, and causes chronic and persistent hepatocyte injury. Chronic hepatic inflammation leads to accelerated hepatocyte apoptosis, necrosis, and regeneration, resulting in recurrent DNA damage, genomic instability, accumulation of mutation, and eventually, neoplastic transformation. Moreover, the overall immunosuppressive environment created for limiting host damage in CHB, including the exhaustion of CD8 effector T cells, creates a tumor-prone immune-microenvironment, which promotes immune evasion during hepatocarcinogenesis. (Figure 1)

### 3.2. CD4+ T Cells

#### 3.2.1. Th1 and Th17 Cells

Naive CD4+ T cells differentiate into various effector T cell subsets. Each subset produces specific cytokines, facilitating different types of immune responses. Th1 cells produce pro-inflammatory cytokines such as IFN-γ, TNF-ɑ, and IL-2 to promote cell-mediated immunity through the induction of the activation of CD8+ T cells. Th1 cells play an essential role in antitumor immunity, as well as antiviral immunity, by producing IFN-γ [48]. The frequency of Th1 cells in HCC tissue in HBV-related HCC was lower than that in the corresponding nontumor tissues, and a small number of Th1 cells was significantly correlated with poor disease-free survival [49]. Th17 cells are characterized by the production of IL-17, IL-21, IL-22, IL-26, and TNF-α. The balance in circulating helper T cells shifts to Th17 dominance from Th1 dominance during chronic HBV infection. As Th17 cells are associated with a high viral load and significant liver injury, this shift is associated with more advanced liver disease and poor prognosis in CHB-related liver disease [50]. Moreover, a higher number of tumor-infiltrating Th17 cells is associated with poor prognosis in patients with HBV-related HCC. A high tumoral Th17-to-Th1 ratio was reported to be an independent predictor of poor disease-free survival and overall survival in patients with HBV-related HCC who underwent liver resection [49].

#### 3.2.2. CD4+ Cytotoxic T Cells

CD4+ cytotoxic T cells secrete granzyme and perforin and exert direct lytic activity dependent on the direct recognition of target cells via major histocompatibility complex class II receptors. Circulating CD4+ cytotoxic T cells are detected in patients with CHB, whereas few CD4+ cytotoxic T cells are detected in healthy individuals. CD4+ cytotoxic T cells exhibit antitumor activities in several malignancies, including HCC [51,52,53]. In HBV-related HCC, CD4+ cytotoxic T cell was progressively deficit as HCC progressed, and decreased number and/or functional impairment of CD4+ cytotoxic T cells was associated with poor clinical outcomes [54]. A higher number of Treg cells was associated with the deficit in CD4+ cytotoxic T cells in patients with HBV-related HCC [54].

### 3.3. Regulatory T Cells

Treg cells are an immunosuppressive subset of CD4+ T cells. They are characterized by the expression of FOXP3 [55,56]. Treg cells play an essential role in maintaining immune tolerance and controlling excessive immune activation after infection with various pathogens [57,58,59]. Treg cells are classified according to their developmental mechanisms into natural Treg (nTreg) cells and peripheral-induced Treg (iTreg) cells or peripherally derived Treg cells [60,61,62,63]. Several studies have demonstrated a higher frequency of Treg cells in patients with CHB than in spontaneously recovered or healthy individuals [64,65]. A significant positive correlation was observed between the frequency of circulating Treg cells and the serum HBV load. These results indicate that Treg cells expand during chronic HBV infection [66,67]. Previous studies demonstrated that persistent exposure to a high level of HBsAg induced the expansion of monocytic MDSCs [68]. MDSCs express and secrete high levels of TGF-β and IL-10. Activated HSCs could also secrete TGF-β. TGF-β and IL-10 promote peripheral conversion of CD4+ T cells into iTreg cells and induce the recruitment of Treg cells [68,69,70,71].

Several mechanisms of antitumor immune suppression by Treg cells have been reported [72,73] as follows: (1) secretion of immunosuppressive cytokines (TGF-β, IL-10, and IL-35), (2) direct killing of effector cells or dendritic cells by cytolysis via granzyme and perforin, (3) direct suppression of target cells via the interaction of CTLA4 on Treg cells and CD80/CD86 with T effector cells or dendritic cells (DCs), and (4) apoptosis of conventional T cells by high IL-2 consumption of Treg cells and depriving existing T cells of IL-2. Thus, Treg cells create a tumor-prone immune microenvironment for HCC formation by weakening the immune surveillance function of the innate and adaptive immune systems (Figure 2).

In HBV-related HCC, the increase in Treg cells suppresses HBV antigen-specific immune responses as well as antitumor immune responses in HCC [74,75]. Previous studies reported that the circulating and tumor-infiltrating Treg cells were correlated with the clinicopathological features associated with poor prognosis, such as tumor vascular invasion, advanced tumor stage, and absence of tumor encapsulation [76]. Moreover, the increased frequency of tumor-infiltrating Treg cells has been reported to be an independent predictor of poor clinical outcomes in patients with HCC [76,77,78,79,80]. These results suggest that the therapeutic strategies for controlling Treg cells may be beneficial for patients with HCC.

### 3.4. B Cells

The role and importance of T cells in tumor immunology are well established by several studies. However, the role of B cells, a major component of adaptive immunity and a major player in humoral immunity, has not been studied in detail in hepatocarcinogenesis. At the onset of HBV infection, B cells mediate the antiviral response by mainly producing antibodies such as anti-HBs and anti-HBc [27]. Moreover, B cells play a role in activating T cell-mediated responses by presenting HBcAg to T cells [81]. Moreover, effector B cells, a distinct subset of B cells, secrete pro-inflammatory cytokines such as IL-6 and IFN-γ, to influence the differentiation of CD4+ T cells into Th1 [82]. HBV-specific B cells present an atypical memory phenotype (CD21-CD27-) in patients with CHB with increased PD-1 expression, resulting in impaired anti-HBs antibody production [83]. The lack of B-cell antibody-producing function leads to failure of HBV clearance and eventually leads to persistent inflammation, which fosters liver disease progression [82].

In chronic inflammatory conditions such as CHB, B cells can differentiate into another distinct subset, regulatory B cells (Breg cell), which increase the production of immunosuppressive cytokines such as IL-10 and TGF-β and present inhibitory molecules such as PD-1 [84,85,86]. Breg cells orchestrate an immunosuppressive environment to limit tissue damage in chronic inflammatory conditions; however, it eventually contributes to tumor evasion by the immune surveillance system [84,87]. In patients with HBV-related HCC, IL-10-expressing B cells preferentially presented with TIM-1 expression, and their frequency was negatively correlated with the frequencies of granzyme A, granzyme B, and perforin-expressing CD4+ T cells [88]. Moreover, the abundance of Breg cells in the tumor margins of the HCC or circulating Breg cells was associated with HCC progression [89]. Therefore, the absence of B cell antibody-producing function and increased proportion of Breg cells in CHB contribute to HBV-induced hepatocarcinogenesis.

## 4. Regulation among Different Immune Cells in HBV-Related HCC

The innate and adaptive immunity function together detect and eliminate transformed cells. The development of HCC may be caused by the immune imbalance induced by HBV infection. The immune system fails to eliminate the HBV during persistent infection, resulting in progressive liver damage and the possibility of HCC development. A study showed that immune–inflammatory interactions between tumors and hosts, i.e., both the innate and adaptive immune responses, play a role in cancer progression [90].

Multiple factors from immune cells play a role in the cross-talk and immune balance in HBV-related HCC (Figure 3). During chronic HBV infection, CD4+T and CD8+T cells, NK cells, NKT cells, monocytes/macrophages, and HSCs induce inflammation, leading to the development of HCC. Furthermore, pro-inflammatory cytokines aggravate the inflammation in chronic hepatitis [16,19,20]. Conversely, immunosuppressive cells, including Treg, Breg, MDSC, and Kupffer cells, produce TGF-β and IL-10 and may act as negative regulators of immune function [5,25,88,91,92].

Excessive activation of immune suppressor cells can lead to persistent HBV infection and progression of HCC. The immunosuppressive effect of cells involved in negative immune regulation causes CD8+ T and NK cells to become exhausted, allowing HBV and HCC tumor cells to escape the immune system [22,72,93,94]. The key immunosuppressive T cell subset, Treg cells, mainly produces immunosuppressive cytokines including TGF-β, IL-10, and IL-35 [72]. According to a meta-analysis, high levels of Treg expression are apparently associated with the development of HCC [91]. Xu et al. showed that Kupffer cell-derived IL-10 inhibited the production of protective antibody of B cells in HBV-persistent mice [21].

Dendritic cells (DCs) are antigen-presenting cells that play a crucial role in host immune responses. A recent study found that plasmacytoid DC function was greatly reduced in patients with CHB, and IL6ST (gp130) played a key role in contributing to DC dysfunction [92]. Another study showed that HBV causes abnormal NK–DC interplay that may impair the efficacy of antiviral immune responses in patients with CHB [93].

## 5. Sex-Related Differences in the Immune System

Previous studies have revealed a sex disparity in liver diseases associated with HBV, which could be a result of sex hormones. Sex hormones are likely to interact with HBV directly and indirectly through their effects on the life cycle, immune response, and progression of liver diseases associated with HBV infection [94]. Consequently, females are able to clear antigens associated with HBV faster than males are, and are better able to control and prevent the progression of HBV-induced liver diseases. The role of HBV as a sex hormone-responsive virus helps explain the sex disparity in HBV-related end-stage liver disease and HCC development.

## 6. Conclusions

Chronic HBV infection induces immune dysregulation in several ways that contribute to HCC development. Despite the existing knowledge on the role of immune cells and related cytokines in HCC, the immunologic and pathogenic aspects that potentially influence the development of HBV-related HCC remain poorly understood.

The development of novel immunotherapeutics will greatly benefit from a deeper understanding of the mechanisms that govern HBV-related HCC immunobiology and pathogenesis. We hope that this review will contribute to current research into the immunopathogenesis of HBV-related HCC.

## Figures and Tables

**Figure 1 ijms-22-08011-f001:**
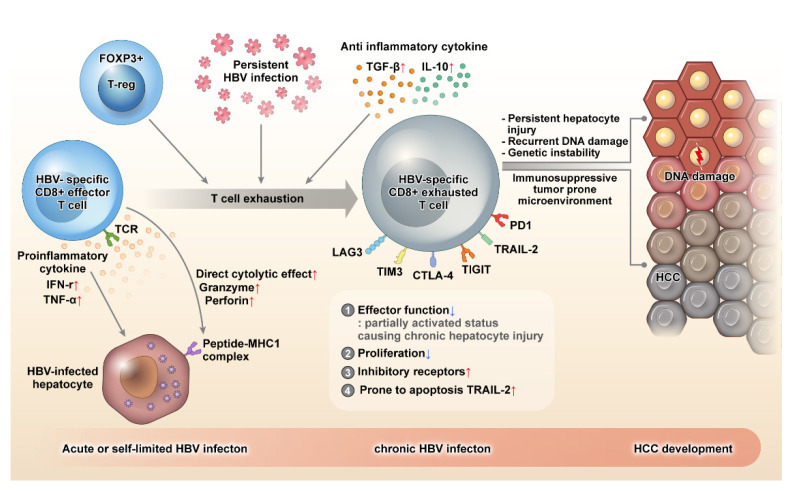
During acute or self-limiting HBV infection, HBV-specific CD8+ effector T cells can control HBV infection effectively by producing pro-inflammatory cytokines, such as IFN-γ and TNF-ɑ, and direct killing of HBV-infected hepatocytes through perforin and granzyme. HBV-specific CD8+ T cells become exhausted in the event of persistent HBV infection. Exhausted CD8+ T cells demonstrate decreased proliferation and are prone to apoptosis by the upregulation of the death receptor TRAIL-2. Although the effector function of CD8+ T cell decreases when these cells are exhausted, a partially activated status is maintained, which causes persistent hepatocyte injury, recurrent DNA damage, genomic instability, mutation accumulation, and eventually, neoplastic transformation. Moreover, exhausted HBV-specific CD8+ T cells express several inhibitory molecules, such as PD-1, TIGIT, CTLA-4, TIM3, and LAG3, and the immune surveillance of CD8+ T cells is weakened, and a tumor-prone immune microenvironment is created. As a result, hepatocytes with neoplastic transformation cannot be removed effectively, and the resulting immune evasion leads to hepatocarcinogenesis. HBV: hepatitis B virus, TIGIT: T cell immunoreceptor with Ig and ITIM domains, PD-1: programmed cell death protein-1, TNF: tumor necrosis factor.

**Figure 2 ijms-22-08011-f002:**
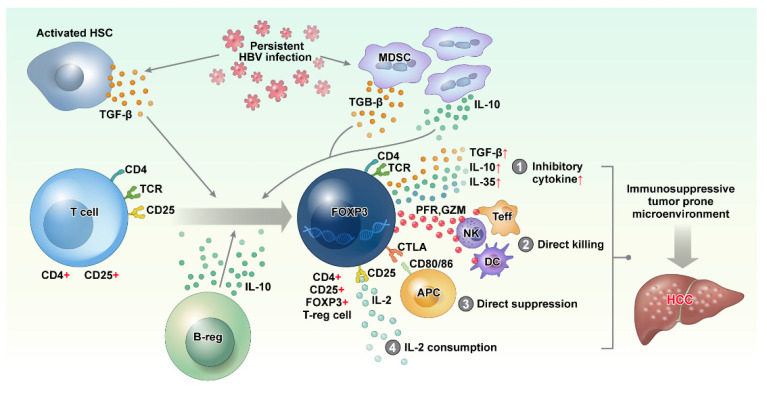
Persistent HBV infection activates hepatic stellate cells (HSCs) and recruits myeloid-derived suppressor cells (MDSC), resulting in the increase in immune-suppressive cytokines, such as TGF-β and IL-10. These immunosuppressive cytokines cause the differentiation of CD4+ CD25+ T cells into CD4+ CD25+ FOXP3 regulatory T cells (Treg cell). Although the function of Treg cells is essential for limiting host damage, persistent HBV infection causes Treg cells to contribute to a tumor-prone immune microenvironment by (1) producing inhibitory cytokines, (2) direct killing of effector cells via perforin and granzyme, (3) direct suppression of antigen presentation cells via CTLA4–CD80/86 interaction, and (4) IL-2 consumption via CD25. HBV: hepatitis B virus, IL: interleukin, TGF: transforming growth factor.

**Figure 3 ijms-22-08011-f003:**
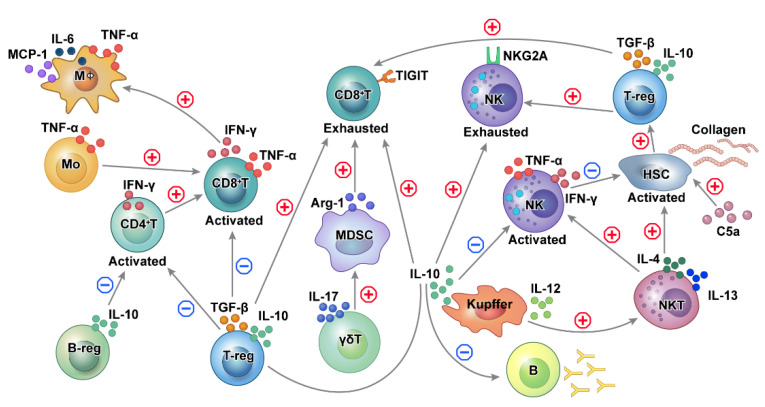
Schematic outline of the immune cell network in HBV-related HCC. Cross-talk and interaction exist among different immune cells in regulating functions promoting hepatocarcinogenesis in HBV-related HCC. In chronic HBV-induced inflammation, activation of adaptive immune cells functions as part of the pro-inflammatory process, while tolerance and exhaustion of adaptive immune cells reduce the detection and elimination of transformed cells. CD8+ T cells are involved in both liver damage and the detection of premalignant liver cells. T cell-mediated tumor surveillance is diminished as a result of Treg recruitment. Kupffer cells promote the activation of NKT cells and NK cells, which activate hepatic stellate cells. Activated CD4+ T, CD8+ T, and NK cells are inhibited by suppressive Tregs, Kuppel cells, and Bregs. HBV: hepatitis B virus, HCC: NK: natural killer cells, NKT cells: natural killer T cells, Treg: regulatory T cells, Breg: regulatory B cells.

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
