# Peer review of "Role of Immune Cells in Patients with Hepatitis B Virus-Related Hepatocellular Carcinoma"

_ijms, 2021, doi:10.3390/ijms22158011_

Round 1

Reviewer 1 Report

The review, well written, focuses on the most innovative basic steps in immunologic field, that mediate carcinogenesis induced by chronic HBV infection.

Please to correct page 2 line 38 hepatocarcinogenesis

Author Response

Manuscript ID: ijms-1300980

Title: Role of immune cells in patients with hepatitis B virus–related hepatocellular carcinoma

We thank you for your thorough checking of our paper. We have checked all presented data and have corrected the text according to the reviewers’ comments and suggestions.

Corrected contents are highlighted by red color

Reviewer: 1 

The review, well written, focuses on the most innovative basic steps in immunologic field, that mediate carcinogenesis induced by chronic HBV infection. Please to correct page 2 line 38 hepatocarcinogenesis

: We corrected typographical error (page 2 line 38, hepatocarcinogenesis)

Sincerely yours

Jae Youn Cheong, M.D

Department of Gastroenterology, Ajou University School of Medicine

16499, 164 Worldcup-ro, Youngtong-ku, Suwon, South Korea

Fax : (82)-31-219-5999

Telephone : (82)-31-219-5119

e-mail : jaeyoun620@gmail.com

Reviewer 2 Report

Title: Role of immune cells in patients with hepatitis B virus-related hepatocellular carcinoma

In this manuscript, by reviewing recent publications, the authors provide extensive information about immune dysregulation induced during acute and chronic HBV infection and about chronic inflammation which contributes to HCC development. It is an interesting and well-written review.  

  • Line 35 “development oof HCC” should be replaced by “development of HCC”
  • 1.2. TOX is not a receptor as is written in manuscript: line 178 -180 : “Exhausted HBV-specific CD8+ T cells characteristically express multiple co-inhibitory receptors, including the programmed cell death protein-1 (PD-1), cytotoxic T-lymphocyte antigen 4 (CTLA-4), T cell immunoglobulin domain and mucin domain 3 (TIM-3), TIGIT, and thymocyte selection-associated high mobility group box (TOX) [43-46]. TOX is the transcription factor! This needs to be corrected even in figure 1 as TOX is not expressed on the surface of exhausted CD8 T cells but intracellularly.
  • Line 249 “These results indicate that the expanded Treg cells during chronic HBV infection [68, 69].” It seems that the verb is missing.
  • The role of dendritic cells, which orchestrate the immune response to HBV, could be more described in the manuscript.
  • The manuscript would benefit from a short paragraph summarizing the sex-related differences in the immune system during HBV infection. A better appreciation of these differences between the two sexes might be of value for a better understanding of the mechanisms that govern HBV-related HCC immunobiology and pathogenesis. (https://doi.org/10.1007/s00281-018-0727-4 ; https://doi.org/10.1155/2017/3214917)

Author Response

Manuscript ID: ijms-1300980

Title: Role of immune cells in patients with hepatitis B virus–related hepatocellular carcinoma

We thank you for your thorough checking of our paper. We have checked all presented data and have corrected the text according to the reviewers’ comments and suggestions.

Corrected contents are highlighted by red color

Reviewer: 2 

In this manuscript, by reviewing recent publications, the authors provide extensive information about immune dysregulation induced during acute and chronic HBV infection and about chronic inflammation which contributes to HCC development. It is an interesting and well-written review. 

1) Line 35 “development oof HCC” should be replaced by “development of HCC”

: We corrected the typographical error (oofà of)

2) 1.2. TOX is not a receptor as is written in manuscript: line 178 -180 : “Exhausted HBV-specific CD8+ T cells characteristically express multiple co-inhibitory receptors, including the programmed cell death protein-1 (PD-1), cytotoxic T-lymphocyte antigen 4 (CTLA-4), T cell immunoglobulin domain and mucin domain 3 (TIM-3), TIGIT, and thymocyte selection-associated high mobility group box (TOX) [43-46]. TOX is the transcription factor! This needs to be corrected even in figure 1 as TOX is not expressed on the surface of exhausted CD8 T cells but intracellularly.

: We removed TOX in the manuscript and figure 1

3) Line 249 “These results indicate that the expanded Treg cells during chronic HBV infection [68, 69].” It seems that the verb is missing.

: We replaced the above sentence as follows:

“These results indicate that Treg cells expand during chronic HBV infection”

4) The role of dendritic cells, which orchestrate the immune response to HBV, could be more described in the manuscript.

: We added the below paragraph in the manuscript

“Dendritic cells (DCs) are antigen-presenting cells that play a crucial role in host immune responses. A recent study found that plasmacytoid DC function was greatly reduced in patients with CHB, and IL6ST (gp130) played a key role in contributing to DC dysfunction [96]. Another study showed that HBV causes abnormal NK-DC interplay that may impair the efficacy of antiviral immune responses in patients with CHB [97]”

5) The manuscript would benefit from a short paragraph summarizing the sex-related differences in the immune system during HBV infection. A better appreciation of these differences between the two sexes might be of value for a better understanding of the mechanisms that govern HBV-related HCC immunobiology and pathogenesis. (https://doi.org/10.1007/s00281-018-0727-4; https://doi.org/10.1155/2017/3214917)

: We added the below paragraph in the manuscript

“5. Sex-related differences in the immune system

Previous studies have revealed a sex disparity in liver diseases associated with HBV, which could be a result of sex hormones. Sex hormones are likely to interact with HBV directly and indirectly through their effects on the life cycle, immune response, and progression of liver diseases associated with HBV infection [98]. Consequently, females are able to clear antigens associated with HBV faster than males, as well as to control and prevent the progression of HBV-induced liver diseases. The role of HBV as a sex hormone-responsive virus helps explain the sex disparity in HBV-related end-stage liver disease and HCC development”

6) Additionally, we corrected the typographical error in figure 2 and replaced it with a revised figure.

Thank you again for your interest in our study and for your patience. If in your opinion other corrections are required, we will be pleased to do so and eagerly await your response.

Sincerely yours

Jae Youn Cheong, M.D

Department of Gastroenterology, Ajou University School of Medicine

16499, 164 Worldcup-ro, Youngtong-ku, Suwon, South Korea

Fax : (82)-31-219-5999

Telephone : (82)-31-219-5119

e-mail : jaeyoun620@gmail.com